# The Neural Contributions to Reactive Balance Control: A Scoping Review of EEG, fNIRS, MRI, and PET Studies

**DOI:** 10.3390/brainsci15121330

**Published:** 2025-12-13

**Authors:** Andrew S. Monaghan, Taylor Takla, Edward Ofori, Daniel S. Peterson, Wendy Wu, Nora E. Fritz, Jason K. Longhurst

**Affiliations:** 1School of Psychology, Queen’s University Belfast, 18-30 Malone Road, David Keir Building, Belfast BT9 5BN, UK; a.monaghan@qub.ac.uk; 2College of Pharmacy and Health Sciences, Wayne State University, 259 Mack Avenue, Detroit, MI 48201, USA; taylortakla@wayne.edu (T.T.); wendy.wu@wayne.edu (W.W.); 3College of Health Solutions, Arizona State University, 550 N. 3rd St., Phoenix, AZ 85004, USA; edward.ofori@asu.edu (E.O.); daniel.peterson1@asu.edu (D.S.P.); 4Doisy College of Health Sciences, Saint Louis University, 3437 Caroline Street, St. Louis, MO 63104, USA

**Keywords:** automatic postural responses, reactive balance, electroencephalography, near-infrared spectroscopy, magnetic resonance, positron emission tomography

## Abstract

**Background/Objectives**: Rapid postural reactions are critical for preventing falls, yet the neural systems supporting these responses are not fully understood, particularly with respect to aging and neurological disorders. Understanding how the brain detects, interprets, and responds to balance disturbances is essential for developing new interventions. This scoping review aimed to synthesize evidence from neuroimaging studies to identify the cortical and subcortical mechanisms underlying reactive balance and to characterize how these mechanisms are altered by aging and pathology. **Methods**: A structured search of EMBASE, PubMed, and CINAHL (7 November 2024) identified studies examining neural activity during experimentally induced balance perturbations. Sixty-one studies met inclusion criteria (EEG n = 45; MRI n = 9; fNIRS n = 8; PET n = 1) and were analyzed for patterns of regional activation and age- or disease-related differences. **Results**: Evidence converges on a distributed network supporting reactive balance. Sensorimotor, premotor, supplementary motor, and prefrontal cortices show consistent involvement, while cerebellar, brainstem, and basal ganglia structures contribute to rapid, automatic responses. Aging and neurological conditions commonly heighten cortical activation, suggesting reduced automaticity and increased reliance on compensatory control. **Conclusions**: Reactive balance emerges from coordinated activity across cortico-subcortical systems that are altered by aging and pathology. Further research incorporating multimodal imaging approaches and more ecologically realistic perturbation paradigms is needed to clarify mechanistic pathways and inform precision-based fall-prevention strategies.

## 1. Introduction

Falls are ubiquitous and have become a major public health concern for communities around the world [1]. Alarmingly, as many as one in three individuals over the age of 65 falls annually, and this number increases among individuals with chronic conditions, such as neurological injury or disease [2,3]. Additionally, falls bring with them a host of negative consequences, including injury, mortality [1], post-fall anxiety syndromes, and activity restriction [2,4]. While falls and fall injuries increase with age, younger individuals are not immune to falls or their consequences [4]. Taken together, the management of falls and fall risk is paramount. Balance plays a prominent role in falls [2] and consists of several sub-domains [5,6]. For example, prior to a balance-challenging task, individuals may release an “anticipatory postural adjustment” to prepare for the impending loss of balance [5,6]. On the other hand, after the task, a reactive or “automatic postural response” may be released to prevent a loss of balance [5,6]. While many aspects of balance control are needed for safe engagement in daily activities, reactive balance deficits strongly contribute to falls and their downstream consequences [7,8]. This is of particular importance in high fall risk populations such as older adults and those with neurological pathology. Despite this, uncertainty remains regarding many of the neural contributions to reactive balance responses [9]. Clarifying the neural mechanisms underlying reactive balance is essential for informing the development of more effective assessment tools and rehabilitation strategies aimed at reducing falls.

The brain regions and networks that facilitate reactive balance responses have been studied frequently over the past two decades [9,10]. Work from human and animal studies indicates the involvement of the brainstem in this response [10,11,12]. More recently, behavioral and neurophysiological evidence has demonstrated involvement of cortical regions [9]. However, the specific mechanisms through which these neural substrates contribute to reactive balance control have not been well described [9]. Additionally, subcortical brain regions with potential involvement appear to be poorly described, such as the thalamus, cerebellum, and basal ganglia structures. Moreover, the neural control of reactive balance responses may be influenced by aging, neurologic disease, or injury [13,14,15,16]. Once more, however, the precise nature of the influence of aging or a specific neurological pathology on reactive balance control remains unclear.

There has been considerable interest in the neural control of reactive balance, evidenced by a recent systematic review on the topic [9]. This review provided insights into the current evidence for cortical involvement in reactive balance. However, it was limited in scope, focusing primarily on mobile neuroimaging approaches, such as electroencephalography (EEG) and functional near-infrared spectroscopy (fNIRS) [9]. Since that time, several additional studies have been conducted, including several focused specifically on factors that influence the neural control for this task. As such, the purpose of this study was to systematically review the current state of the literature regarding the neural contributions to reactive balance control. Specifically, we aim to provide clarity for the following questions: (1) Which cortical and subcortical structures are involved in reactive balance responses, and how do they interact to facilitate reactive balance control? (2) Does aging influence the neural control of reactive balance? (3) Do the neural contributions to reactive balance change with neurologic pathologies? (4) What methodological challenges exist that limit the current literature in this area? Additionally, this review sought to understand each of these questions separately across different imaging modalities: EEG, fNIRS, magnetic resonance (MR) imaging, and positron emission tomography (PET). Addressing these questions will help identify potential therapeutic targets for interventions aimed at improving reactive balance control, ultimately contributing to reduced fall risk.

## 2. Materials and Methods

The reporting for this systematic review was guided by the five steps to conducting a systematic review [17] and followed the guidelines for Preferred Reporting Items for Systematic Reviews and Meta-Analyses extension for Scoping Reviews (PRISMA-ScR) [18]. Given the intention of this review to map the literature across a broad topic, a scoping review was selected [19,20]. As the purpose of this review was to explore the general scope of research conducted on the topic, quality appraisal of studies was not performed. An a priori protocol was designed in consultation with a research librarian, and the review was registered on Open Science Framework [21].

Literature searches were conducted in EMBASE, PubMed, and CINAHL to identify relevant studies from the inception of databases to 7 December 2024. Search algorithms had the following format: automatic postural response AND neuroimaging. Keywords, along with their associated medical subject headings (MeSH), terms, and CINAHL complete subject headings, were used as appropriate to identify relevant articles. A complete search algorithm for PubMed is given as an example in Appendix A. Reference lists of relevant studies were manually reviewed to identify additional articles not captured by our database searches. Search duplicates were removed using Covidence systematic review software (Veritas Health Innovation, Melbourne, Australia).

Searches were designed to include all possible relevant studies that might contribute data for the review but exclude letters, editorials, comments, case reports, historical articles, reports, protocols, withdrawals, retractions of publications, retracted publications, replies, published errata, conference abstracts, dissertations, book chapters, conference proceedings, and studies on children or animals only using database built-in filters. Non-English articles were also excluded from the searches due to constraints on translation capability. All study types of adults with both a neuroimaging outcome and exposure to an acute unexpected balance challenge or an unpredictable postural perturbation were included. Studies including only visual perturbations were excluded. Studies that tested an intervention were included only if they presented baseline neuroimaging and reactive balance data.

A total of 4124 studies underwent two rounds of screening using Covidence based on their (1) title and abstract and (2) full text. Each round of screening, eligibility assessment, and data extraction was performed by independent investigators (JL, AM, NF, TT, EO, DP), with any discrepancies resolved through discussion and consensus. Sixty-one articles, including data obtained from EEG, MRI, fNIRS, and PET imaging, were selected for data extraction and analysis using Excel. The number of included and excluded studies is reported in the PRISMA (the Preferred Reporting Items for Systematic Reviews and Meta-Analysis) (Figure 1).

Data was extracted manually and charted using Microsoft Excel. Data elements included the following: citation, sample characteristics, sample size, imaging modality, approaches for application of imaging modality (e.g., coordinate system and location of electrodes for EEG), characteristics of analyses, description of data processing procedures, perturbation characteristics (e.g., type and direction), imaging results related to perturbation or reactive postural control, and included brain regions. Results were organized and presented by imaging modality (EEG, fNIRS, MRI, and PET). Within each modality, the studies were summarized regarding the following topics: the findings related directly to reactive balance generally, the findings related to aging and neurologic disease, and methodological approaches for EEG and fNIRS. Critical appraisal of the quality of each of the articles was not undertaken in this review. Results were summarized narratively and in tabular format.

## 3. Results

Out of 5903 studies identified, 1779 duplicates were identified, leaving a total of 4124 unique articles that were identified by the search strategy as potentially relevant to the purpose of this study. During title and abstract screening, 3929 of these were identified as not applicable to this review and were excluded. A total of 195 studies underwent full-text review, of which 61 were deemed relevant and included in this review. Of the 61 articles included, 45 included data obtained via EEG, nine included data obtained via MRI, eight included data obtained via fNIRS, and one article included PET imaging. Two articles included data obtained from multiple imaging modalities.

### 3.1. EEG Overview

EEG was used to assess neural response to reactive balance paradigms in 45 included studies (Table 1). Among those studies, 39 included healthy young adults (HYA) [13,16,22,23,24,25,26,27,28,29,30,31,32,33,34,35,36,37,38,39,40,41,42,43,44,45,46,47,48,49,50,51,52,53,54,55,56,57,58], ten included healthy older adults (HOA) [13,27,44,50,58,59,60,61,62,63], five included samples of individuals with neurologic injury or disease (traumatic brain injury (TBI), n = 2; Parkinson’s disease (PD), n = 1; cerebrovascular accident (CVA), n = 2) [16,24,30,60,62], one study included a sample of individuals with high fall risk [59], and one included young adults with low back pain [64]. The direction of perturbations delivered varied across 7 studies that administered perturbations in multiple directions [13,16,22,23,59,60,64]; 34 studies delivered perturbations in the sagittal plane (anterior–posterior) only [13,24,25,26,27,28,29,30,31,32,33,34,35,38,39,40,41,42,43,44,47,48,49,50,51,52,53,54,55,56,57,58,61,62,63], 4 studies utilized only coronal plane (medial-lateral) perturbations [36,37,45,46], and 2 utilized pitch perturbations [13,64]. Other perturbation parameters (amplitudes, accelerations, and velocities, etc.) were not consistent across included studies. The type of EEG analysis approaches also varied among studies; 18 utilized event-related potential (ERP) analysis [22,23,27,31,32,39,40,41,42,43,44,47,48,60,62,63,64], 8 employed spectral/time-frequency (including event-related spectral perturbation) analysis [25,35,53,55,57,58,59], 6 used independent component analysis (ICA) decompositions [16,24,33,38,51,52], and 8 used connectivity analyses [13,30,36,45,50,54,56,61]. Additionally, five studies included source localization approaches [26,29,34,37,46].

#### 3.1.1. EEG Findings Related to Reactive Balance

Studies using ERP analysis consistently noted reactive balance-related activity in fronto-central cortical regions, particularly the supplementary motor area (SMA), which may reflect early sensory processing or error detection of postural perturbations, as indexed by the N1 component. N1 amplitude and latency are influenced by attentional state, predictability, and task relevance [22,23,31,39,41,42]. P2 responses, although less frequently examined, were linked to SMA and parietal sources and may reflect later-stage performance monitoring [40,64]. Spectral analyses consistently reveal theta synchronization across midfrontal and parietal regions during balance challenges, in line with action monitoring and postural adaptation [36,46,52,53]. Alpha desynchronization and beta rebound fluctuate with cognitive load and threat level [50,59], supporting the notion that reactive balance engages a distributed cortical network involving sensorimotor, parietal, and prefrontal regions [25,28,50,51].

#### 3.1.2. Influence of Aging and Neurologic Injury or Disease for EEG Analyses

Aging and neurologic disease alter both the timing and topography of EEG responses to postural perturbations. Older adults show delayed and attenuated early ERPs (P1, N1) during perturbations, consistent with slowed sensory processing [27,63]. Spectral studies in HOA report increased frontal theta and parieto-occipital beta/gamma power scaling with challenge, reflecting compensatory recruitment [50,59]. In TBI survivors, N1 amplitude is reduced [24], and functional connectivity is disrupted [30], indicating network segregation deficits. PD alters N1 metrics compared to age-matched peers [62]. Participants post-CVA exhibit smaller, delayed N1 responses [16,60], and participants with chronic low back pain (LBP) show enlarged P2 amplitudes with delayed postural muscle onsets [64].

#### 3.1.3. EEG Methodologic Approaches

Artifact contamination from eye movements and postural muscle activity presents a significant challenge; several studies utilize ICA combined with artifact subspace reconstruction for artifact rejection [30,51,52]. Scalp-based source localization provides cortical mapping but lacks the resolution for deeper structures [34,46]. Small, cross-sectional cohorts, sometimes as few as three to four patients [16,35], limit the generalizability of findings. Most research employs tightly controlled support-surface translations; only a few studies implement more naturalistic perturbations, such as virtual reality [59] or unexpected treadmill slips [35]. The heterogeneity in perturbation types, EEG montages, and analysis pipelines further complicates direct comparisons. Future advancements, such as high-density EEG with individualized MR source models [29], standardized preprocessing procedures, and real-world paradigms, are expected to improve spatial precision and translational relevance.

### 3.2. fNIRS Overview

Eight studies were included that utilized fNIRS to assess neural response to a postural perturbation (Table 2) [14,65,66,67,68,69,70,71]. Among those studies, five included HYA [14,65,66,67,68], one study included HOA [14], and three studies included samples of individuals with neurologic injury or disease (PD n = 1, CVA n = 2) [69,70,71]. In regard to perturbation parameters, seven studies employed support surface translations, one of these while walking [65], while one study utilized an anterior mechanical pull [14]. The direction of the perturbation varied, with six studies delivering anterior perturbations [14,66,68,69,70,71], five delivering posterior perturbations [65,66,68,70,71], one delivering coronal plane perturbations [68], and one study utilizing a rotary perturbation [67]. While six studies designed perturbations to require in-place automatic postural responses [66,67,68,69,70,71], two studies designed perturbations to require stepping responses [14,65]. Specific perturbation parameters (amplitudes, accelerations, and velocities, etc.) were not consistent across studies.

#### 3.2.1. fNIRS Findings Related to Reactive Balance

Across studies, perturbations consistently elicited increased activation in prefrontal regions [14,65,66,68,69,70,71]. Several studies reported that this activation was strongest during the initial perturbations and diminished with repeated exposures. An exception was the orbitofrontal cortex, where activation remained stable over time [65]. Additionally, increases in activation were observed in the SMA [66,70], premotor cortex (PMC) [14,66,71], primary motor cortex [66], and superior parietal cortex [66,71]. No pattern of hypoactivation with perturbation was reported across any of the studies. One study noted increased PFC activation associated with transcranial direct current stimulation during perturbation compared to perturbation with sham stimulation [69]. In contrast to the other studies, one study found no difference in the hemodynamic signal during the period during and immediately following perturbation compared to the rest condition [67].

#### 3.2.2. Influence of Aging and Neurologic Injury or Disease for fNIRS Analyses

Age-related differences were examined in one study, noting that PFC and PMC activation associated with perturbations was increased in older adults compared to young adults [14]. Another study found that individuals with PD exhibited increased PFC activation [69], consistent with patterns observed in other neurologic groups. Two additional studies involving individuals post-CVA reported increased activation in the PFC [70,71], SMA [70], PMC [71], and superior parietal cortex activation [71] in response to perturbations. Notably, only one of these post-CVA studies identified hemispheric differences in activation patterns [71]. None of the three studies that included neurologic populations included a comparison group.

### 3.2.3. fNIRS Methodologic Approaches

To better understand the methodological challenges and opportunities in the utilization of perturbation paradigms and reactive balance research, data regarding fNIRS methodologies were extracted. Among the seven studies that utilized fNIRS, there was a wide variety in both cortical regions observed and methodologies utilized for data processing. All studies utilized montages that included portions of the frontal lobe; however, five also included parietal regions [14,66,68,70,71], while two studies utilized montages that were restricted to only the PFC [65,69]. Considering the recently published guidelines for the use of fNIRS in posture and gait research [72], the review team also examined the methods for artifact removal, data processing approach, and outcome measures reported. Regarding artifact removal, there was very little consistency in methods reported among the studies (see Table 2). Additionally, one study reported no information regarding artifact removal [67]. Notably, only one study [68] utilized short separation channels, which are a powerful tool to remove extracerebral hemodynamic artifacts [72]. Lastly, all studies reported only oxygenated hemoglobin as the primary outcome.

### 3.3. MR Overview

Nine studies were included, which utilized magnetic resonance imaging (MRI) to assess neural response to a postural perturbation (Table 3) [15,30,73,74,75,76,77,78]. Among those studies, one included HYA [73], two studies included HOA [15,74], and eight studies included samples of individuals with neurologic injury or disease (PD n = 2; TBI n = 1; multiple sclerosis (MS) n = 2; mild cognitive impairment (MCI) n = 2) [15,30,74,75,76,77,78]. In regard to perturbation parameters, six studies employed support surface translations, one of these while walking [73], and five while the participant was standing [30,75,76,77,78]. Two studies used a push-and-release paradigm [15,74]. Seven studies examined cross-sectional relationships among MRI and reactive balance [15,30,74,75,76,77,78], while one study delivered slip perturbation training programs and examined pre/post differences [73]. The direction of the perturbation varied, with five studies delivering anterior perturbations [30,73,76,77,78] and five delivering posterior perturbations [15,30,74,75,76]. While one study designed perturbations to require in-place automatic postural responses [77], five studies designed perturbations to require stepping responses [15,73,74,75,78], and two did not specify a response [30,76]. Specific perturbation parameters (amplitudes, accelerations, velocities, etc.) were not consistent across studies.

#### 3.3.1. MR Findings Related to Reactive Balance

##### fMRI

Only one study examined task-based fMRI [73]. In this study, HYA performed 3 days of treadmill-slip perturbation training. Pre- and post-intervention, participants underwent a task-based fMRI with an imagined (i.e., mental imagery) slipping and imagined walking paradigm. At baseline, compared to rest, imagined slipping resulted in increased activation in frontal, parietal, and limbic regions, including the superior frontal gyrus (SMA and BA6), inferior frontal gyrus, inferior parietal lobule, parahippocampal gyrus, cingulate gyrus, and posterior cerebellum. After 3 days of training, imagined slipping resulted in increased activation of frontal, parietal, occipital, and lingual areas (Table 3). There were also significant differences between imagined slipping and imagined walking, with imagined slipping resulting in increased activation of cerebellar, temporal, and frontal areas, including SMA and parahippocampal gyrus (Table 3). Notably, fMRI was not directly compared to behavioral performance on the perturbation task, which perhaps explains the large number of activated brain areas observed during imagined slipping [73].

##### Brain Volume

Two studies examined whole brain volume [74,75]. In older persons [74], better postural control, as measured by step height, was associated with greater volume in brainstem, cerebellar, and parietal areas, which aligns with work in older adults with MCI [75], demonstrating that better postural control, as measured by postural center of mass, was linked with higher gray matter volume in brainstem and cerebellar areas.

##### Diffusion Tensor Imaging (DTI) Connectivity

Four studies examined DTI connectivity [30,75,76,77]. Across studies, poorer stepping [75,79] and longer latencies in the tibialis anterior [76,77] with perturbations were associated with lower fractional anisotropy (FA) in the spinal cord [76] and brainstem (pedunculopontine nucleus) [77] regions, as well as corticospinal [75], corticostriatal [75], corticothalamic [75], frontopontine [75], and parietopontine tracts [75]. Interestingly, Handiru et al. found no relationship between center of pressure displacement to platform perturbations at a small amplitude and global DTI metrics [30]. Finally, Monaghan and colleagues assessed relationships between MR and changes in reactive stepping through training. Although associations with baseline stepping were not run (and therefore this study was not included in Table 3 above), results suggested that retention of step length increases following perturbation training was associated with higher FA in anterior thalamic radiation, posterior thalamic radiation, superior longitudinal fasciculus, and inferior longitudinal fasciculus [80].

##### Resting State

Two studies examined resting state [15,78]. Among persons with PD, cerebellar connectivity was related to reactive balance, as measured by step length. This relationship was not observed in age-matched control subjects [15]. Similar findings were also observed in individuals with MCI, suggesting cerebello–cortical connectivity is associated with greater reactive balance stability [78]. The finding that brainstem/cerebellar connectivity was related to reactive balance is generally consistent with the brainstem and cerebellar contributions noted in human studies utilizing brain volume and DTI (see Sections Brain Volume and Diffusion Tensor Imaging (DTI) Connectivity) and animal literature [81,82]. It is notable that the relevant cerebellar connectivity was with auditory and visual regions in people with PD [15]. The role of the cerebellar vermis in integrating sensory inputs could provide some rationale for this finding [15].

#### 3.3.2. Influence of Aging and Neurologic Injury or Disease for MR Analyses

Several investigations included neurological populations [15,74,76,77] and some also included analyses of control groups [15,74]. However, while these studies identified some key findings regarding neural control of reactive balance, none of these studies directly examined age-related or pathology-related differences in reactive balance control. As such, there is little evidence on whether age or neurologic injury/disease impacts the neural mechanisms of reactive balance.

### 3.4. PET

Only one study was included, which utilized PET imaging to assess neural response to a postural perturbation (see Table 3) [79]. The association between striatal dopamine denervation, related to aging, and reactive balance response to slip perturbations during gait was investigated among a sample of 50 adults ranging from 20 to 86 years of age in this cross-sectional study. Slip perturbations were achieved during gait by having participants walk across a floor that was unexpectedly contaminated with glycerol solution. Greater dopamine denervation in the caudate nucleus, but not the putamen, was associated with a greater magnitude of slipping, leading to the conclusion that age-related striatal dopamine depletion may impact the ability to recover from a large perturbation during walking. Additionally, the involvement of the caudate specifically may implicate the frontostriatal pathways involved in the executive control of gait when large perturbations challenge postural control [83,84]. However, this result was only present among participants who walked fast (greater than 1.2 m/s). The authors postulate that the slow walkers may have adopted cautious gait behavior to limit their exposure to fall risk and thus blunt the relationship with striatal dopamine denervation. This study provided initial evidence of the influences of the dopaminergic system and aging on reactive balance control, but further evidence is needed to understand these relationships more fully.

## 4. Discussion

This review sought to understand the current state of literature regarding the neural contributions to reactive balance control. The results of this review suggest that reactive balance is influenced by a multifaceted processing involving cortical and subcortical brain regions; see Figure 2. Further, both aging and neurologic pathologies likely influence the neural mechanisms that underlie reactive balance control. Additionally, several methodological challenges were identified, and recommendations for future investigations were observed. Notably, the results of this review expand upon the findings from the previous review by Purohit and Bhatt [9] in several notable ways. While their work focused primarily on mobile neuroimaging (EEG and fNIRS) and cortical responses, our synthesis incorporates evidence from MRI and PET studies, revealing structural and neurochemical contributions of subcortical regions such as the cerebellum, brainstem, and basal ganglia. This broader scope highlights that reactive balance is not solely a cortical phenomenon but emerges from integrated cortico-subcortical networks. Additionally, we examined aging and neurological conditions alongside healthy young adults, identifying compensatory cortical recruitment and connectivity changes that were not addressed previously. Finally, this review emphasizes methodological challenges and opportunities, including artifact removal, ecological validity, and the need for multimodal imaging to advance mechanistic understanding.

### 4.1. Regional Neural Control of Reactive Balance

#### 4.1.1. Cortical Involvement in Reactive Balance

The findings from this review demonstrate that reactive balance control engages a broad network of cortical regions, underscoring the importance of cerebral cortical processing in response to postural perturbations (see Figure 2). The frontal cortex, particularly the PFC and SMA, emerged as the most consistently involved regions across imaging modalities, with additional contributions from parietal and, to a lesser extent, occipital areas. EEG studies have provided robust evidence for cortical involvement through ERP components such as the N1 and P2. The N1 potential, typically originating from frontal, premotor, and supplementary motor regions [38], reflects rapid sensory processing of balance perturbations and is sensitive to factors like threat, predictability, and task demands. The P2 component, often localized to the anterior cingulate cortex, is thought to reflect performance monitoring and error processing during postural corrections. Beyond these findings, broader prefrontal involvement has also been implicated. Specifically, the dorsolateral PFC has been observed during balance recovery [65,66,68]. This has also been observed during other challenging balance tasks, particularly when those tasks impose higher cognitive demands, suggesting that these regions may contribute to executive control and potentially a compensatory role of cognitive control resources when short-latency automatic balance responses are insufficient alone [85,86].

Beyond ERPs, time-frequency has revealed increased theta and beta band power following balance disturbances [46,53,54,55]. These oscillatory dynamics are proposed to support sensorimotor integration and attentional control, particularly under conditions requiring flexible postural response patterns [46,87]. fNIRS studies further support these findings, showing increased hemodynamic responses in the PFC and SMA during perturbations [14,66,68,69,71], with some studies indicating modulation based on exposure or task repetition [65,70], suggesting a role in learning and adaptation.

Collectively, these results suggest that cortical regions are not only involved in high-level modulation of balance but may also be engaged in medium-to-long latency responses to destabilizing events. This is consistent with contemporary understanding of neural recruitment during balance corrections [10]. Evidence suggests that short latency responses are facilitated via activation of a mono- or oligosynaptic spinal circuit [10,88,89]. Whereas medium and long latency responses likely arise from the brainstem, which communicates sensory information to the cortex, particularly the SMA, PFC, and PMC, and other subcortical regions (basal ganglia and cerebellum) [10,82,90]. While the short latency response may not involve the cortex, evidence suggests that cortical loops are likely involved in shaping and adapting postural responses [10,91]. The evidence presented for cortical involvement supports an integrated model of reactive balance control in which cortical sensory, motor, and association areas dynamically interact to evaluate threat, contextual demands, and task relevance in order to guide appropriate balance corrections [9,10,92]. These insights have important implications for understanding age-related and disease-related changes in balance control, where increased cortical recruitment may reflect compensatory mechanisms in response to diminished subcortical or peripheral efficiency.

#### 4.1.2. Subcortical Involvement in Reactive Balance

Subcortical structures, including the brainstem, cerebellum, basal ganglia, and thalamus, are active during reactive balance control, particularly in the rapid integration of sensory inputs and initiation of motor responses following postural perturbations. Structural MRI and DTI studies from this review consistently demonstrate associations between subcortical integrity and reactive balance performance, providing correlational evidence for their involvement in this task [78,79]. For example, decreased fractional anisotropy, indicative of poorer microstructural integrity, in the pedunculopontine nucleus, a key brainstem region implicated in locomotor and postural regulation, was associated with delayed muscle response latencies in persons with MS, suggesting impaired timing of automatic postural responses [77]. Similarly, in individuals with MCI, diminished structural integrity in corticostriatal, corticospinal, corticothalamic, and frontopontine tracts was correlated with poorer postural control [75], suggesting a role for projection fibers in effective balance reactions.

The cerebellum has also been linked to reactive balance, with several studies reporting that reduced cerebellar volume or connectivity is associated with poorer reactive stepping or postural control. In HOA, larger cerebellar and brainstem volumes were associated with greater reactive step height [74], an indicator of effective postural correction. Resting-state fMRI further demonstrated that cerebellar connectivity with visual and auditory networks was predictive of reactive balance performance in individuals with PD [15,73], potentially reflecting the cerebellum’s role in integrating sensory inputs and guiding adaptive responses. Additionally, PET imaging findings indicated that dopamine denervation in the caudate nucleus was associated with poorer reactive balance in older adults, implicating basal ganglia circuits in the modulation of rapid postural corrections during dynamic tasks such as gait [80]. Taken together, these findings reinforce the notion that reactive balance control relies not only on cortical contributions but also critically on the structural and functional integrity of subcortical systems.

#### 4.1.3. Cortical and Subcortical Interaction in Reactive Balance

Findings from this review suggest that reactive balance control relies on coordinated activity between cortical and subcortical regions. Cortical areas, including the PFC and SMA, consistently show increased activation during postural perturbations, particularly under conditions requiring cognitive engagement or adaptation [52,61,66,68,70,71]. At the same time, subcortical structures such as the cerebellum, brainstem, and basal ganglia are implicated in the medium latency components of balance responses, with multiple studies showing that reduced structural integrity or connectivity in these regions is associated with delayed or less effective motor outputs [75,77]. Notably, multimodal findings from Handiru and colleagues demonstrated that reduced white matter integrity was associated with altered cortical functional connectivity during balance tasks, suggesting that subcortical pathways may directly influence cortical network organization and efficiency [30]. Together, these findings support the concept of a distributed cortico-subcortical network in which subcortical structures provide fast, automatic responses, while cortical regions may be recruited for modulation, adaptation, or compensation.

### 4.2. The Influence of Aging on Neural Control of Reactive Balance

There is limited evidence regarding age-related changes to the neural control of balance responses following perturbations. Only three of the included studies included analyses comparing older and younger adults on perturbation-related neuroimaging measures (EEG [13,50] and fNIRS [14]). Taken together, the findings of these studies indicate that as individuals age, there is a greater neural response to loss of balance in the PFC and PMC [14], and perhaps across the area spanning the frontoparietal region [13,50]. These increases in neural activation appear to be associated with increased cognitive engagement and utilization of attentional resources for balance correction [93,94]. One possible explanation of this finding is that aging results in decreased speed, efficiency, and consistency of recruitment of postural responses and the associated motor units [95]. In this case, the increase in neural and cognitive resources associated with a loss of balance may be a compensatory response to maintain appropriate and timely balance responses [96]. This approach is consistent with previous work examining brain activity during walking across age and neurological conditions [97,98]. Spectral EEG findings further support this hypothesis, showing reduced alpha and beta power in older adults, indicative of impaired sensorimotor integration and altered cortical dynamics following perturbations [50]. However, due to the small number of studies directly comparing age groups and the variability in experimental designs, these findings should be interpreted with caution. Clarifying how aging may impact neural mechanisms supporting reactive balance will be essential for informing preventative and rehabilitative approaches to reduce falls in older populations.

### 4.3. The Influence of Neurologic Pathology on Neural Control of Reactive Balance

There remains limited evidence of the impact of different neurological pathologies on the neural correlates of automatic postural responses. This review found few articles that compared neural responses to balance perturbations between individuals with neurologic pathology and healthy controls [15,16,24,30,60,62,74]. Among these, differences in cortical and subcortical involvement were observed across populations with PD, stroke, TBI, and MCI. For example, in people with PD, but not non-PD older adult peers, resting-state cerebellar connectivity was associated with reactive stepping responses [15], suggesting disease-specific alterations in cerebellar contributions to postural control. In stroke survivors, N1 responses are slower and smaller and associated with balance performance [60], potentially reflecting delayed sensorimotor error detection, reduced cortical excitability, and impaired ability to quickly engage neural resources that contribute to balance recovery. In TBI, reduced N1 amplitudes and weakened EEG functional connectivity during perturbations may indicate impaired sensory processing and reduced network efficiency [30]. Additionally, individuals with MCI exhibited reduced structural integrity in subcortical white matter tracts (e.g., corticospinal and corticostriatal) associated with diminished postural control, as measured by center of mass displacement [75]. Although these findings collectively support the notion that neurologic pathology alters both cortical and subcortical contributions to reactive balance, the small number of studies, heterogeneity in pathology types, and methodological variability limit definitive conclusions. However, these results do underscore the broad regions that contribute to reactive balance. Further research with larger samples and inclusion of individuals with neurological pathologies is necessary to delineate the unique and shared neural mechanisms underlying balance impairments across neurologic conditions.

### 4.4. Methodologic Challenges

Investigating the neural correlates of reactive balance using neuroimaging techniques presents numerous methodological challenges that limit consistency, comparability, and generalizability across studies. Some of these challenges reflect inherent strengths and limitations of each modality, while others arise from methodological choices that could be standardized or refined. One major issue lies in the variability related to experimental design, such as differences in perturbation parameters, task demands, and response requirements, which complicate the synthesis of findings across studies. These design inconsistencies are further compounded by the inherent strengths and limitations of each imaging modality. EEG provides excellent temporal resolution to capture rapid cortical responses, but is inherently vulnerable to contamination from eye and muscle artifacts and has poor spatial resolution. Careful artifact removal and standardized task paradigms could help mitigate these influences. fNIRS offers fair to good spatial resolution in addition to portability, supporting its ecological validity for reactive balance tasks. However, fNIRS has limited penetration depth, is sensitive to extracerebral hemodynamics, and has limited temporal specificity, which may be insufficient for certain applications regarding activation sequences associated with rapid, discrete behaviors (such as reactive balance). Notably, adopting consistent preprocessing procedures and incorporating short-separation channels can improve data quality. MR approaches provide high spatial resolution and can be applied across the entire brain; however, MR possesses poor temporal resolution, and, in contrast to EEG and fNIRS, balance tasks cannot be conducted during MR acquisition, which hinders interpretations. These limitations can be partially mitigated by study design choices and employing multimodal imaging to fill inherent gaps. PET imaging uniquely captures neurochemical contributions to reactive balance but is inherently constrained by low temporal resolution [79]. Moving forward, addressing these methodological challenges requires standardization of experimental paradigms, consistent artifact removal methods and data processing strategies, and larger, more diverse patient samples across all imaging modalities [72,99]. Additionally, multimodal imaging approaches can help mitigate the challenges associated with a particular imaging modality. Overcoming these challenges is essential for advancing our understanding of the neural control of reactive balance and improving clinical applications in balance rehabilitation and fall prevention.

### 4.5. Clinical Recommendations

The growing evidence of cortical and subcortical involvement in medium- and long-latency reactive balance responses suggests that reactive postural control relies on virtually all aspects of the neural axis, and different aspects of the postural response can be influenced by cognitive, attentional, and executive processes. This highlights the potential value of dual-task assessments and interventions that challenge both cognitive and motor systems simultaneously as potential tools to assess and train reactive balance. Interventions targeting frontal network engagement, such as task-specific training, balance under cognitive load, or neuromodulation, may be particularly relevant for older adults or individuals with neurologic conditions. Moreover, clinicians should be aware that increased cortical recruitment observed in aging and disease may reflect compensatory mechanisms, and, thus, balance deficits in these populations may not be due solely to physical deconditioning but also to neural inefficiency or reorganization. Finally, it is notable that most of the work on this topic has focused on cortical activation due to methodological challenges of assessing contributions of deep brain structures (particularly the brainstem) to reactive balance. However, as noted above, seminal work in animal studies underscores the contributions of these structures in reactive balance [82,88,89,90,91]. Further, clinical approaches exist to directly target deep brain structures (e.g., deep brain stimulation, DBS). While the effects of DBS on balance, in general, are varied [100,101], effects on reactive balance in particular are less common and require additional investigation [101,102].

### 4.6. Limitations

This scoping review has several limitations that should be acknowledged. While we aimed for comprehensive coverage, the search strategy was restricted to articles published in English, potentially introducing language bias and excluding relevant studies in other languages. Additionally, the review did not formally assess the methodological quality or risk of bias of included studies; therefore, the strength of evidence cannot be fully evaluated. Lastly, the included studies were conducted in controlled laboratory settings, which may not adequately represent real-world reactive balance scenarios.

## 5. Conclusions

This review underscores the complexity of neural control underlying reactive balance, revealing contributions from both cortical and subcortical systems. In healthy young adults, the most frequently studied population, reactive balance engages a distributed cortical network, including the PFC, SMA, and parietal regions, alongside subcortical structures such as the cerebellum and brainstem. EEG and fNIRS findings indicate that these cortical regions support rapid sensory integration and adaptive control, while oscillatory dynamics (e.g., theta and beta power) reflect flexible engagement of attentional and motor processes. MR evidence further highlights the role of subcortical integrity in shaping automatic responses. These results suggest that even in young adults, reactive balance is not purely automatic but involves higher-order cortical contributions. Across modalities, aging and neurological pathology are associated with heightened cortical activation and disrupted network efficiency, suggesting compensatory recruitment. However, methodological heterogeneity and limited ecological validity remain significant barriers. Standardized protocols and multimodal imaging approaches are essential to advance mechanistic understanding and translate findings into targeted interventions that enhance reactive balance for fall-risk reduction across the lifespan.

## Figures and Tables

**Figure 1 brainsci-15-01330-f001:**
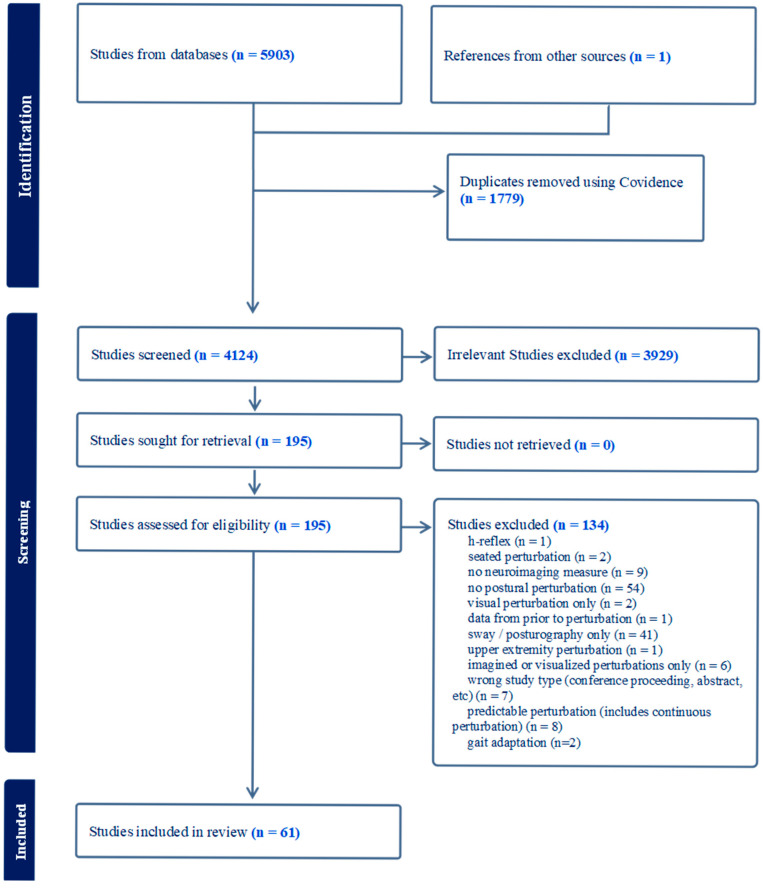
PRISMA diagram.

**Figure 2 brainsci-15-01330-f002:**
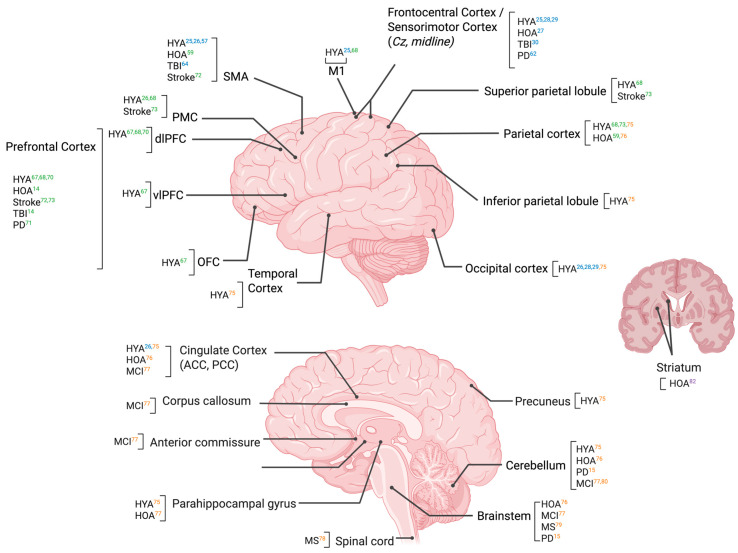
Neural regions supporting reactive balance control. The numbers shown in the figure correspond to the sequence numbers of the references listed in the caption. Color indicates the imaging modality providing evidence for each region: blue (EEG) = cortical electrophysiological activity (ERPs, oscillations); green (fNIRS) = hemodynamic activation in frontal and parietal cortices; orange (MRI) = structural and connectivity correlates in cortical–subcortical networks; and purple (PET) = dopaminergic contributions to reactive balance. Abbreviations: PFC = prefrontal cortex; SMA = supplementary motor area; PMC = premotor cortex; M1 = primary motor cortex; SPL/IPL = superior/inferior parietal lobule; ACC = anterior cingulate cortex. ***References***: [14] R. Zhuang, S. Zhu, Y. Sui, M. Zhou, T. Yang, C. Wang, T. Zhang, J. Wang, C. Kan, Y. Shen, T. Wang, C. Guo, Age-Related Differences in Stepping Reactions to a Balance Perturbation: A Functional Near-Infrared Spectroscopy and Surface Electromyography Study, *Brain Sci* **2022**, *12(11)*. [15] A. Ragothaman, M. Mancini, J.G. Nutt, D.A. Fair, O. Miranda-Dominguez, F.B. Horak, Resting state functional networks predict different aspects of postural control in Parkinson’s disease, *Gait & Posture* **2022**, *97*, 122-129. [25] S. Boebinger, A. Payne, G. Martino, K. Kerr, J. Mirdamadi, J.L. McKay, M. Borich, L. Ting, Precise cortical contributions to sensorimotor feedback control during reactive balance, *PLoS Comput Biol* **2024**, *20(4)*, e1011562. [26] M.D. Bogost, P.I. Burgos, C.E. Little, M.H. Woollacott, B.H. Dalton, Electrocortical Sources Related to Whole-Body Surface Translations during a Single- and Dual-Task Paradigm, *Front Hum Neurosci* **2016**, *10*, 524. [27] R.B. Duckrow, K. Abu-Hasaballah, R. Whipple, L. Wolfson, Stance perturbation-evoked potentials in old people with poor gait and balance, *Clin Neurophysiol* **1999**, *110(12)*, 2026-32. [28] N.J. Ghosn, J.A. Palmer, M.R. Borich, L.H. Ting, A.M. Payne, Cortical Beta Oscillatory Activity Evoked during Reactive Balance Recovery Scales with Perturbation Difficulty and Individual Balance Ability, *Brain Sci* **2020**, *10(11)*. [29] R. Goel, R.A. Ozdemir, S. Nakagome, J.L. Contreras-Vidal, W.H. Paloski, P.J. Parikh, Effects of speed and direction of perturbation on electroencephalographic and balance responses, *Exp Brain Res* **2018**, *236(7)* 2073-2083. [30] V. Shenoy Handiru, A. Alivar, A. Hoxha, S. Saleh, E.S. Suviseshamuthu, G.H. Yue, D. Allexandre, Graph-theoretical analysis of EEG functional connectivity during balance perturbation in traumatic brain injury: A pilot study, *Hum Brain Mapp* **2021**, *42(14)*, 4427-4447. [57] J.P. Varghese, A. Marlin, K.B. Beyer, W.R. Staines, G. Mochizuki, W.E. McIlroy, Frequency characteristics of cortical activity associated with perturbations to upright stability, *Neurosci Lett* **2014**, *578*, 33-8. [59] C.J. Chang, T.F. Yang, S.W. Yang, J.S. Chern, Cortical Modulation of Motor Control Biofeedback among the Elderly with High Fall Risk during a Posture Perturbation Task with Augmented Reality, *Front Aging Neurosci* **2016**, *8*, 80. [62] A.M. Payne, J.L. McKay, L.H. Ting, The cortical N1 response to balance perturbation is associated with balance and cognitive function in different ways between older adults with and without Parkinson’s disease, *Cereb Cortex Commun* **2022**, *3(3)*. [24] D. Allexandre, A. Hoxha, V.S. Handiru, S. Saleh, S.E. Selvan, G.H. Yue, Altered Cortical and Postural Response to Balance Perturbation in Traumatic Brain Injury - An EEG Pilot Study, *Annu Int Conf IEEE Eng Med Biol Soc* **2019**, 1543-1546. [65] B.C. Lee, J. Choi, B.J. Martin, Roles of the prefrontal cortex in learning to time the onset of pre-existing motor programs, *PLoS One* **2020**, *15(11)*, e0241562. [66] M. Mihara, I. Miyai, M. Hatakenaka, K. Kubota, S. Sakoda, Role of the prefrontal cortex in human balance control, *Neuroimage* **2008**, *43(2)*, 329-36. [68] R. Nishimoto, S. Fujiwara, Y. Kutoku, T. Ogata, M. Mihara, Effect of dual-task interaction combining postural and visual perturbations on cortical activity and postural control ability, *NeuroImage* **2023**, 120352. [69] V.S. Beretta, R. Vitório, P. Nóbrega-Sousa, N.R. Conceição, D. Orcioli-Silva, M.P. Pereira, L.T.B. Gobbi, Effect of Different Intensities of Transcranial Direct Current Stimulation on Postural Response to External Perturbation in Patients With Parkinson’s Disease, *Neurorehabil Neural Repair* **2020**, *34(11)*, 1009-1019. [70] H. Fujimoto, M. Mihara, N. Hattori, M. Hatakenaka, T. Kawano, H. Yagura, I. Miyai, H. Mochizuki, Cortical changes underlying balance recovery in patients with hemiplegic stroke, *Neuroimage* **2014**, *85(1)*, 547-54. [71] M. Mihara, I. Miyai, N. Hattori, M. Hatakenaka, H. Yagura, T. Kawano, K. Kubota, Cortical control of postural balance in patients with hemiplegic stroke, *Neuroreport* **2012**, *23(5)*, 314-9. [73] P.J. Patel, T. Bhatt, S.R. DelDonno, S.A. Langenecker, S. Dusane, Examining Neural Plasticity for Slip-Perturbation Training: An fMRI Study, *Front Neurol* **2018**, *9*, 1181. [74] A. Ragothaman, O. Miranda-Dominguez, B.H. Brumbach, A. Giritharan, D.A. Fair, J.G. Nutt, M. Mancini, F.B. Horak, Relationship Between Brain Volumes and Objective Balance and Gait Measures in Parkinson’s Disease, *J Parkinsons Dis* **2022**, *12(1)*, 283-294. [75] L. Kannan, T. Bhatt, A. Zhang, O. Ajilore, Association of balance control mechanisms with brain structural integrity in older adults with mild cognitive impairment, *Neurosci Lett* **2022**, *783*, 136699. [76] C.Y. Lee, J.M. Huisinga, I.Y. Choi, S.G. Lynch, P. Lee, Correlation between spinal cord diffusion tensor imaging and postural response latencies in persons with multiple sclerosis: A pilot study, *Magn Reson Imaging* **2020**, *66*, 226-231. [77] D.S. Peterson, G. Gera, F.B. Horak, B.W. Fling, Supraspinal control of automatic postural responses in people with multiple sclerosis, *Gait Posture* **2016**, *47*, 92-5. [78] L. Kannan, T. Bhatt, O. Ajilore, Cerebello-cortical functional connectivity may regulate reactive balance control in older adults with mild cognitive impairment, *Front Neurol* **2023**, *14*, 1041434. [80] R. Cham, S. Perera, S.A. Studenski, N.I. Bohnen, Age-related striatal dopaminergic denervation and severity of a slip perturbation, *J Gerontol A Biol Sci Med Sci* **2011**, *66(9)*, 980-5.

**Table 1 brainsci-15-01330-t001:** Electroencephalography (EEG) studies.

Authors	Population	EEG Characteristics	Type of EEG Analysis	Type of Perturbation	Perturbation Direction	Main Findings	Brain Regions
Adkin et al., 2008 [23]	HYA (n = 10)	Cz electrode	ERP (N1 amplitude)	Mechanical trunk push (in-place)	Multidirectional	(1) N1 larger under high postural threat (2) N1 modulation correlated with fear and confidence ratings	Fronto-central cortex (Cz)
Adkin et al., 2006 [22]	HYA (n = 8)	Midline electrodes (Cz, FCz, CPz), linked mastoids	ERP (N1 amplitude and latency)	Mechanical trunk push (in-place)	Multidirectional	(1) Reduced N1 for predictable perturbations (2) N1 returned during surprise trial	Fronto-central cortex (Cz, FCz)
Allexandre et al., 2019 [24]	TBI (n = 12), HYA (n = 6)	64-channel EEG, ICA, DIPFIT	ERP (N1), PEP analysis	Support surface translations (in-place)	Anterior–posterior	(1) TBI group showed lower N1 amplitude vs. controls (2) N1 amplitude and BBS correlated; N1 latency related to greater CoP displacement	SMA (Brodmann area 6), midline sensorimotor cortex
Boebinger et al., 2024 [25]	HYA (n = 17)	32-channel EEG (Cz-focused), ICA	ERP (N1), β power (13–30 Hz), SRM modeling	Support surface translations (in-place)	Posterior	(1) Cortical N1 and β activity time-locked to perturbations and scale with difficulty (2) SRM models show cortical signals predict later balance-correcting EMG	SMA, primary motor cortex, sensorimotor cortex (Cz)
Bogost et al., 2016 [26]	HYA (n = 15)	256-channel EEG, ICA, DIPFIT	ERP (N1), Source localization (single vs. dual task)	Support surface translations (in-place)	Posterior	(1) Dual-task reduced N1 amplitude (2) Cortical source shift from sensorimotor regions to temporal and occipital areas during dual-task	SMA, PMC, somatosensory, anterior cingulate, occipital cortex
Chang et al., 2016 [59]	HOA, high and low fall risk (n = 31)	32-channel EEG, PSD (theta, alpha, beta, gamma)	Spectral analysis (VR vs. non-VR)	Continuous dynamic perturbations with VR	Multidirectional	(1) High fall risk group showed less coordinated cortical response during the visual–vestibular challenge (2) Beta/gamma increased in parietal–occipital; theta increased in frontal–central regions during challenge	Parietal–occipital (β, γ), frontal–central (θ), occipital (α)
Duckrow et al., 1999 [27]	HYA (n = 8), HOA (n = 33)	13-channel EEG (10–20), Cz-focused	ERP (P1, N1, N2), inter-peak interval	Support surface translations	Anterior	(1) Older adults showed delayed and bifid N1–N2 complex (2) Inter-peak N1–N2 latency longer in mobility-impaired elders and correlated with balance performance	Midline vertex (Cz), frontoparietal cortex
Ghosn et al., 2020 [28]	HYA (n = 19)	32-channel EEG, Cz wavelet ERSP	Spectral (β power 13–30 Hz), time–frequency	Support surface translations (in-place)	Posterior	(1) β power increased with perturbation magnitude (2) Late β (150–250 ms) scaled with poorer balance ability	Sensorimotor cortex (Cz)
Goel et al., 2018 [29]	HYA (n = 10)	64-channel EEG, AMICA ICA, DIPFIT	ERP (N1 latency and amplitude), IC source localization	Support surface translations (in-place)	Anterior–posterior	(1) N1 latency shorter for forward perturbations; amplitude modulated by speed (2) Postural response latency followed N1 latency; source localized to fronto-central IC	Fronto-central cortex, sensorimotor cortex
Handiru et al., 2021 [30]	TBI (n = 17), HYA (n = 15)	64-channel EEG, ASR, ICA, DIPFIT, source-localized to 68 ROIs	Functional connectivity, graph-theory metrics (modularity, efficiency)	Support surface translations (in-place)	Posterior	(1) TBI group showed reduced alpha-band segregation and beta-band connectivity (2) Theta-band modularity negatively correlated with balance performance (BBS)	Sensorimotor, occipital, frontal regions
Jacobs et al., 2016 [64]	Chronic LBP (n = 13), controls (n = 13)	128-channel EEG, focus on midline electrodes	ERP (N1, P2); P2 as marker of late cortical processing	Platform rotations (toes-up, toes-down)	Bidirectional	(1) LBP group had larger P2 amplitude and delayed trunk muscle onsets (2) P2 amplitude negatively correlated with CoM displacement, pain interference, and fear scores	SMA(P2), Cz
Jacobs et al., 2008 [31]	HYA (n = 12)	Midline electrodes (Cz, Pz, Fz, F3, F4)	ERP (CNV)	Support surface translations (stepping)	Posterior	(1) CNV observed only in cued condition and associated with reduced CoP displacement (2) CNV amplitude correlated with extent of CoP modulation across cue conditions	Sensorimotor cortex; (Cz)
Little & Woollacott 2015 [32]	HYA (n = 14)	Cz and surrounding electrodes, ERP time-locked to perturbation	ERP (N1 amplitude and latency)	Support surface translations during visual working memory dual-task	Posterior	N1 amplitude significantly attenuated in dual-task vs. single-task condition	Motor, sensory, parietal, frontal cortex
Magruder et al., 2024 [33]	HYA (n = 20)	64 channel EEG, ICA, frontocentral focus	Power spectral analysis (delta and alpha), event-related spectral perturbations	Platform translations vs. sway-referenced platform	Anterior–posterior	(1) Delta power higher and alpha power lower in translations than sway-referenced tasks (2) Delta significantly reduced by cTBS over SMA	Frontocentral regions, SMA
Marlin et al., 2014 [34]	HYA (n = 11)	64 channel EEG, ICA, source localization with individual MRIs	ERP (N1), dipole source modeling	Lean-and-release perturbations in standing (in-place)	Anterior	N1 localized to medial frontal gyrus (SMA; BA6), not ACC	N1: SMA, ERN: anterior cingulate
Mezzina et al., 2019 [35]	HYA (n = 4)	13 channel EEG, time-locked to EMG onset	PSD slope (m) in θ, α, β I/II/III bands	Treadmill walking with unexpected slips	Anterior–posterior (slip)	(1) Sharp increase in PSD slope across all bands immediately after perturbation (2) PSD modulation suppressed in recovery step vs. perturbed step	Motor and sensorimotor cortex (C3, Cz, C4)
Mierau et al., 2017 [36]	HYA (n = 37)	32 channel EEG	Directed functional connectivity (theta/alpha bands)	Support surface translations in single limb support (in-place)	Mediolateral	(1) Theta network linked frontal-central-parietal sites; alpha from occipital to parietal (2) Increased theta connectivity and alpha desynchronization during instability	Frontal, central, parietal (θ); occipital → parietal (α)
Mierau et al., 2015 [37]	HYA (n = 37)	32 channel EEG, LORETA	ERP (P1, N1), single-trial analysis, source localization	Support surface translations in single limb support (in-place)	Mediolateral	(1) N1 amplitude adapted over repeated perturbations; correlated with EMG/sway (2) P1 remained stable; N1 localized to SMA, not ACC	P1: BA5 (parietal); N1: BA6 (SMA), BA24 (ACC)
Mirdamadi et al., 2024 [38]	HYA (n = 19)	64 channel EEG, AMICA ICA, DIPFIT, FOOOF, beta burst detection	ERP (N1), spectral (β power), time–frequency	Support surface translations (2AFC perceptual discrimination task)	Posterior (with lateral deviation)	(1) Greater N1 attenuation on correctly perceived trials; associated with better balance (2) Higher pre-perturbation β power and β event metrics associated with worse perception	SMA (N1 source), midline central cortex (β power)
Mochizuki et al., 2017 [39]	HYA (n = 10)	32 channel EEG	Pre- and post-perturbation activity, ERP (N1), area under curve analysis	Lean-and-release perturbations with/without concurrent cognitive task	Anterior	(1) Dual-task reduced pre-perturbation EEG and N1 amplitude under predictable conditions (2) No dual-task effect on N1 under unpredictable perturbations	Fronto-central cortex
Mochizuki et al., 2009 [40]	HYA (n = 8)	64 channel EEG, focusing on FCz and CPz electrodes	ERP (N1, P2), spatio-temporal analysis	Mechanical weight release in standing and sitting	Posterior	N1 amplitude and timing consistent across sitting and standing tasks	Fronto-central cortex
Mochizuki et al., 2008 [41]	HYA (n = 15)	64 channel EEG	ERP (Bereitschaftspotential, N1), DC shifts	Load-release perturbation (self-triggered vs. unpredictable)	Posterior	(1) Predictable perturbations evoked pre-perturbation DC shifts; N1 was smaller and earlier (2) Unpredictable perturbations evoked larger and later N1 responses	SMA, Cz
Ozdemir et al., 2018 [13]	HYA (n = 10), HOA (n = 9)	64 channel EEG, ICA	Time–frequency, cortico-muscular coherence (CMC), ERP (PEP)	Support surface perturbations (translational and rotational)	Anterior, posterior, toes-up, toes-down	(1) Older adults aged over 65 showed elevated gamma, delta power and increased CMC during perturbed stance (2) PEPs showed longer latency and reduced amplitude in HOA vs. young adults	Sensorimotor cortex (Cz, C1, C2), frontal, parietal regions
Palmer et al., 2021 [61]	HOA (n = 16)	64 channel EEG, ICA, beta band (13–30 Hz)	Time–frequency (beta power), coherence (motor–prefrontal/somatosensory)	Support surface translations (in-place)	Anterior	(1) Greater post-perturbation beta power associated with lower balance function (2) Higher motor–PFC coherence linked to increased cognitive interference and lower reactive threshold	Motor cortex (Cz), PFC, somatosensory cortex
Palmer et al., 2025 [60]	CVA (n = 18), HOA (n = 17)	64 channel EEG, ICA, Cz focus	ERP (N1), latency, amplitude	Support surface translations (in-place)	Posterior-lateral, backward	(1) N1 responses are smaller and delayed in CVA group; greatest latency during paretic-leg loading (2) Slower N1 responses associated with poorer clinical balance (MiniBEST, TUG, gait speed) and reduced CoP rate of rise	Cz, sensorimotor cortex (SMA, ACC, parietal)
Payne et al., 2022 [62]	HOA (n = 19), PD (n = 16)	32 channel EEG, ICA, Cz focus	ERP (N1: amplitude, latency, width)	Support surface translation (in-place)	Anterior–posterior	(1) N1 metrics differentially associated with balance and cognitive function across groups (2) In PD: shorter N1 width associated with worse balance/cognition; in HOA: higher amplitude linked to worse cognition/confidence	SMA, ACC, sensorimotor cortex
Payne et al., 2021 [63]	HOA (n = 19)	32 channel EEG, ICA, Cz focus	ERP (N1: amplitude, latency, width)	Support surface translation (in-place)	Anterior–posterior	(1) Poorer set-shifting linked to higher N1 amplitude and stiffer balance recovery (2) Cognitive set shifting deficits associated with low muscle directional specificity and larger cortical response	SMA (Cz), central motor regions
Payne & Ting 2020a [42]	HYA (n = 20)	32 channel EEG, ICA, Cz focus	ERP (N1: amplitude, latency, width)	Support surface translation (in-place) Support surface translations	Posterior	(1) Larger N1 amplitudes associated with worse performance on beam-walking (2) Greater N1 modulation by perturbation size in individuals with lower balance ability	Cz (SMA), frontocentral cortex
Payne & Ting 2020b [43]	HYA (n = 19)	32 channel EEG, ICA, Cz focus	ERP (N1: amplitude, latency, width)	Support surface translation (in-place and stepping)	Posterior	(1) Cortical N1 larger during compensatory stepping vs. non-stepping (2) N1 amplitude not influenced by prior motor planning (planned vs. unplanned steps)	SMA (Cz), central motor regions
Payne et al., 2019 [44]	HYA (n = 15), HOA (n = 18)	32 channel EEG, ICA, Cz/FCz focus	ERP (N1), comparison with ERN	Support surface translations vs. flanker task	Posterior (balance), manual (flanker)	(1) Balance N1 similar in morphology and source to error-related negativity (ERN) (2) Larger N1 amplitudes with unpredictable perturbations and higher threat contexts	SMA, ACC, midline frontal cortex
Peterson & Ferris 2019 [45]	HYA (n = 30)	136 channel EEG, ICA, DIPFIT	Effective connectivity (dDTF), corticomuscular coherence	Waist pull and visual field rotation	Mediolateral	(1) Visual rotation reduced occipito-parietal connectivity (2) Pull increased sensorimotor connectivity and cortico-muscular communication	Occipital-parietal (visual), sensorimotor, SMA, ACC (pull)
Peterson & Ferris 2018 [46]	HYA (n = 30)	136 channel EEG, ICA, DIPFIT	ERSP (θ, β), cluster-based source analysis	Waist pull and visual field rotation during stance and walking	Mediolateral	(1) Both perturbations evoked θ synchronization and β desynchronization (2) Visual rotation evoked occipito-parietal activity; pull evoked sensorimotor activity	Occipital, parietal (visual); sensorimotor (physical)
Quant et al., 2004 [47]	HYA (n = 7)	Single-channel EEG (Cz)	ERP (N1 amplitude and latency, late PEPs)	Support surface translation (in-place)	Anterior	(1) Cognitive dual-task attenuated N1 amplitude (2) Attenuated N1 associated with increased CoP displacement and TA EMG	Cz (vertex), sensorimotor cortex
Quant et al., 2005 [48]	HYA (n = 7)	Single-channel EEG (Cz)	ERP (P1, N1, P2, N2), amplitude/latency comparisons	Support surface translation (in-place)	Posterior	(1) Changes in later postural responses did not alter P2 or N2 potentials (2) Later cortical responses appear independent of motor execution	Vertex (Cz)
Saadat et al., 2021 [50]	HYA (n = 19), HOA (n = 20)	32 channel EEG, coherence	Spectral (alpha, beta power), coherence analysis	Load release perturbations (cable pull, predictable/unpredictable)	Anterior	(1) Older adults had higher beta power and alpha/beta coherence in late postural phase (2) Unpredictable perturbations evoked lower alpha and higher beta power in older adults	Frontal, parietal (F3, F4, P3, P4), sensorimotor (C3, C4)
Sibley et al., 2010 [49]	HYA (n = 10)	64 channel EEG	ERP (N1, P2), electrodermal (EDL, EDR)	Load release perturbation (LOW vs. HIGH platform)	Anterior	(1) N1 amplitude larger at elevated height (HIGH) vs. ground level (2) N1 increase was independent of autonomic arousal (EDR, EDL)	Cz (fronto-central)
Solis-Escalante et al., 2021 [51]	HYA (n = 11)	126 channel EEG, artifact reduction (ASR), ICA	ERP (P1, N1), time-frequency (θ, α, β)	Support surface translations (forward/backward), variable intensity	Anterior–posterior	(1) N1 amplitude and θ/α/β power scaled with perturbation intensity and predicted stepping behavior (2) Cortical responses consistent with action monitoring and increased with higher postural challenge	Midfrontal cortex (SMA, ACC), parietal cortex
Solis-Escalante et al., 2020 [16]	HYA (n = 6), CVA (n = 3)	126 channel EEG, artifact reduction (ASR), ICA	Classification of spectral features (3–50 Hz); theta-focused	Support surface translations (multidirectional sway)	Multidirectional	(1) Cortical theta activity (3–10 Hz) encoded direction-specific postural sway (2) Classification performance similar in stroke and controls	Frontal, central, parietal, occipital
Solis-Escalante et al., 2019 [52]	HYA (n = 10)	126 channel EEG, artifact reduction (ASR), ICA	ERSP (θ, α, β, low-γ), ICA source-resolved	Support surface translation (in-place and stepping)	Posterior	(1) Feet-in-place (high-demand) responses showed greater α and γ suppression in SMA; stepping evoked greater β suppression in contralateral M1/S1 (2) Theta enhanced in PFC and ACC across conditions; stronger in PFC during feet-in-place	SMA, M1/S1, PFC, ACC
Stokkermans et al., 2022 [53]	HYA (n = 15)	126 channel EEG, ICA	Theta power (3–8 Hz), time-locked to foot strike	Multidirectional perturbations inducing reactive steps	Anterior–posterior	(1) Midfrontal theta increased after foot strike, especially during backward stepping (2) Theta correlated positively with margin of stability	Midfrontal cortex
Stokkermans et al., 2023A [54]	HYA (n = 18)	126 channel EEG, ICA	Spectral Granger causality (CMC), time-frequency (1–100 Hz)	Support surface translations (feet-in-place or step)	Anterior–posterior	(1) CMC increased across θ, α, β, and γ bands during step responses (2) No clear relation between CMC and EMG magnitude	Midfrontal cortex
Stokkermans et al., 2023B [55]	HYA (n = 20)	126 channel EEG, ICA	Theta power (3–8 Hz)	Support surface translations (feet-in-place or step)	Anterior–posterior	(1) Midfrontal theta increased with postural threat (e.g., congruent lean and perturbation direction) (2) Theta dynamics indexed internal model updating and balance monitoring	Midfrontal cortex
Varghese et al., 2019 [56]	HYA (n = 19)	64 channel EEG, ICA, source-localized (Desikan atlas)	Functional connectivity, graph theory (MSC, iCoh, ERPC)	Lean-and-release perturbation (feet-in-place)	Anterior	(1) Reactive balance control evokes frequency-specific network reorganization (delta–beta) (2) Increased short-range connections and connection strength during PEP N1	Distributed network: SMA, PMC, sensorimotor, PFC, parietal
Varghese et al., 2014 [57]	HYA (n = 14)	64 channel EEG, ICA, Cz/FCz focus	ERP (N1), power spectral (delta–beta), ERSP, ITC	Lean-and-release perturbation (feet-in-place)	Anterior	(1) Significant N1 response with power increase in delta, theta, alpha, and beta bands (2) Phase-locking in 1–20 Hz bands supports phase reorganization model of ERP generation	Fronto-central cortex (FCz), SMA
Wang et al., 2023 [58]	HYA (n = 20), HOA (n = 20)	10 channel EEG	Time-domain (RMS amplitude), single-trial, sex/age comparisons	Support surface translations (unexpected/predictable)	Anterior–posterior	(1) EEG RMS larger for unpredictable vs. predictable perturbations at Cz and mastoid (2) Older adult males showed greatest RMS response at mastoid; mastoid may be a viable EEG marker	Cz, Fz, mastoid

Cohorts: HYA, healthy young adults; HOA, healthy older adults; PD, Parkinson’s disease; CVA, cerebrovascular accident; TBI, traumatic brain injury; LBP, low back pain. EEG methods: ERP, event-related potential; PSD, power spectral density; ERSP, event-related spectral perturbation; ICA, independent components analysis (e.g., AMICA); ASR, artifact subspace reconstruction; SRM, stimulus-selective response modulation; DIPFIT, dipole fitting; ROI, region of interest; FOOOF, fitting oscillations and one-over-F. Analyses and metrics: CNV, contingent negative variation; PEP, perturbation-evoked potential; CMC, cortico-muscular coherence; dDTF, direct directed transfer function; MSC, magnitude-squared coherence; iCoh, imaginary coherence; ERPC, event-related phase coherence; ITC, inter-trial coherence; 2AFC, two-alternative forced choice; RMS, root-mean-square; VR, virtual reality. Brain regions: SMA, supplementary motor area; M1, primary motor cortex; S1, primary somatosensory cortex; PFC, prefrontal cortex; ACC, anterior cingulate cortex; BA, Brodmann area; PMC, premotor cortex.

**Table 2 brainsci-15-01330-t002:** Functional near-infrared spectroscopy (fNIRS) studies.

Study	Study Paradigm	Group	Perturbation Type	Perturbation Direction	Imaging Finding	Measurement Approach
Beretta et al., 2020 [69]	PFC hemodynamic response evoked by perturbations preceded by tDCS (1 mA or 2 mA) or sham	PD, n = 24, 68.9 ± 8.5 years, 10 females	In-place Surface translation	Posterior platform translation with resultant anterior loss of balance	tDCS associated ↑ in HbO in PFC in both stimulated and non-stimulated hemispheres.	***Montage***: Restricted to PFC ***Software***: Oxysoft and MATLAB, NIRS-SPM ***Data processing***: *Movement*: wavelet detrending *Peripheral Physiology*: low-pass filter, wavelet detrending
Fujimoto et al., 2014 [70]	SMA hemodynamic response evoked by perturbations	CVA, n = 20, 60.2 ± 9.5 years, 3 females	In-place Surface translation	Anterior and posterior	↑ HbO in PFC and SMA associated with better balance performance	***Montage***: Frontal and Parietal regions ***Software***: not reported ***Data processing*:** *Movement*: high-pass filter *Peripheral Physiology*: Principal components analysis and regression of 1st component and constant
Lee et al., 2020 [65]	PFC hemodynamic response evoked by perturbations from static standing and walking	HYA, n = 10, 22.7 ± 3.2 years, 5 females	Stepping via Surface translation and Walking with surface translation Perturbation occurred with left belt only (left lower extremity only)	Anterior surface translation with resultant posterior stepping response	↑ dlPFC, ventrolateral PFC (vlPFC), and orbitofrontal cortex following perturbation during initial 3 perturbations ↓ Activity progressively with perturbation exposure (trial to trial) in the dlPFC, vlPFC, and frontopolar PFC, but orbitofrontal cortex activity ↑ and then remained constant Changes over exposure parallel kinematic improvement in balance recovery	***Montage***: Restricted to PFC ***Software***: MATLAB (specific software not reported) ***Data processing***: *Movement*: movement artifact reduction algorithm *Peripheral Physiology*: Band-pass filter, multi-channel regression
Mihara et al., 2008 [66]	Frontoparietal hemodynamic response evoked by perturbations from static standing	HYA, n = 15, 29.4 ± 6.7 years, 6 females	In-place Surface translation	Anterior and posterior	↑ HbO bilateral middle frontal gyrus (BA 6, 8, 9), ↑ HbO bilateral superior frontal gyrus (BA 8) ↑ HbO right Precentral Gyrus (BA 6) ↑ HbO right superior parietal lobe (BA 7)	***Montage***: Frontal and Parietal regions ***Software***: custom MATLAB program ***Data processing***: *Movement*: high-pass filter *Peripheral Physiology*: none reported
Mihara et al., 2012 [71]	Frontoparietal hemodynamic response evoked by perturbations from static standing	CVA, n = 20, 61.6 ± 11.9 years, 5 females	In-place Surface translation	Anterior and posterior	↑ HbO in bilateral PFC, PMC, and parietal association cortex in unaffected hemisphere	***Montage***: Frontal and Parietal regions ***Software***: not reported ***Data processing***: *Movement*: high-pass filter *Peripheral Physiology*: none reported
Mitsutake et al., 2015 [67]	Frontoparietal hemodynamic response evoked by rotary perturbations from static standing	HYA, n = 12, 25.8 ± 2.1 years	In-place Surface rotation	rotation	HbO levels were unchanged	***Montage***: Frontal regions ***Software***: MATLAB, NIRS-SPM ***Data processing***: *Movement*: none reported *Peripheral Physiology*: none reported
Nishimoto et al., 2023 [68]	Frontoparietal hemodynamic response evoked by perturbations from static standing.	HYA, n = 24, 24.4 ± 5.4 years, 13 females	In-place Surface translation	Anterior, posterior, horizontal	↑ HbO in DLPFC observed across postural perturbation tasks.	***Montage***: Frontal and Parietal regions ***Software***: custom MATLAB program ***Data processing***: *Movement*: none reported *Peripheral Physiology*: short separation channels, but not specified how utilized in analyses
Zhuang et al., 2022 [14]	Frontal hemodynamic response evoked by perturbations from static standing	HYA, n = 20, 27.3 ± 4.9 years, 8 females HOA, n = 20, 68.6 ± 5.5 years, 12 females	Stepping Mechanical Pull	Anterior mechanical pull with resultant anterior stepping response	↑ HbO in PFC and PMC for the HOA group only	***Montage***: Frontal and Parietal regions ***Software***: NirSpark ***Data processing***: *Movement*: spline interpolation *Peripheral Physiology*: band pass filter

HYA, healthy young adults; HOA, healthy older adults; PD, Parkinson’s disease; CVA, cerebrovascular accident; dlPFC, dorsolateral prefrontal cortex; PFC, prefrontal cortex; SMA, supplementary motor area; PMC, premotor cortex; HbO, oxygenated hemoglobin; BA, Brodmann areas; ↑, increased; ↓, decreased.

**Table 3 brainsci-15-01330-t003:** Magnetic resonance (MR) or positron emission tomography (PET) studies.

Study	Imaging Modality	Group	Perturbation Type	Perturbation Direction	Primary Findings	Secondary Findings
Cham et al., 2011 [79]	11c-beta-CFT dopamine transporter PET	Healthy adults ranging from 20 years old to 80 years old—mean age = 64.9 years. (n = 50)	A single slip perturbation induced by walking over an unexpected slippery surface	Not controlled	↓ tracer uptake, indicative of ↑ dopamine denervation, in the caudate nucleus was associated with higher peak slip velocity among fast walkers. No relationship was observed in the putamen or with slow walkers.	
Goel et al., 2018 [29]	*EEG* T1-weighted Structural MRI	Healthy young adults (n = 10)	In-place	Anterior and Posterior at 2 different displacements (3.17 cm, 6.35 cm), 2 different speeds (7.93 cm/s and 15.88 cm/s), and 2 different periods (400 ms, 800 ms)	No MR results reported; MR used to support EEG findings.	
Handiru et al., 2021 [30]	*EEG* DTI-Connectivity T1-weighted Structural	Traumatic brain injury (n = 17)	Not specified	Anterior and Posterior directions; high and low amplitude COM displacement after perturbation Analysis limited to high amplitude in the posterior direction	There were no relationships among DTI global metrics (FA, MD, MA) and behavioral measures of COP or Berg Balance Scale that survived multiple comparisons.	There was a relationship between the structural integrity of the WM system and the strength of the functional connections measured with EEG. The only relationship to survive multiple comparisons: ↑ Beta-band network segregation ~ ↑ global MA
Kannan et al., 2022 [75]	Whole Brain Volumes DTI-Connectivity	Older adults with mild cognitive impairment (n = 10)	Stepping	Forward belt translation; posterior step	**↓ postural COM ~ ↓ FA** ↓ postural COM stability with: ↓ L and R corticostriatal tract ↓ L and R corticospinal tract ↓ L and R corticothalamic tract ↓ L and R frontopontine tract ↓ L and R parietopontine tract ↓ R arcuate faciculus ↓ R cingulum ↓ R inferior longitudinal fasciculus ↓ anterior commissure ↓ corpus callosum (all R > 0.7; all *p* < 0.04) Cerebellar FA was not correlated with reactive balance response.	**↓ postural COM ~ ↓ GM volume** ↓ postural COM stability with: ↓ L and R cerebellar cortex ↓ R accumbens ↓ brainstem (all R > 0.8; all *p* < 0.01)
Kannan et al., 2023 [78]	Resting state fMRI	Older adults with mild cognitive impairment (n = 11)	Stepping in response to support surface translation	Anterior	↑ FC between the cerebellum and frontoparietal, salience, and cerebellar networks was associated with ↑ reactive stability	↓ FC between cerebellum/vermis and sensorimotor and default mode networks was associated with ↑ reactive stability
Lee et al., 2020 [76]	DTI-Connectivity	Relapsing Remitting Multiple Sclerosis (n = 17)	Not specified	Anterior and Posterior	↓ FA of the spinal cord (C4-C6) was significantly correlated with longer latencies measured on the right tibialis anterior in response to forward postural perturbations (r = −0.51, *p* = 0.04)	DTI metrics showed no significant differences between subjects with and without spinal cord lesions. No significant relationships between medial gastrocnemius latency and FA during forward falls (backward perturbations)
Patel et al., 2019 [73]	fMRI-task-based (mental imagery tasks)	Healthy young adults (n = 10)	Stepping	Slip-like perturbation (of increasing intensities) while walking on a treadmill at self-selected speed	At baseline: Compared to rest, imagined slipping resulted in ↑ activation in the frontal, parietal, and limbic regions, including the superior frontal gyrus (SMA, BA6), inferior frontal gyrus, inferior parietal lobule, parahippocampal gyrus, cingulate gyrus, and posterior cerebellum.	After training: ↑ in activation in the left middle frontal gyrus (DLPFC, BA9), right superior parietal lobule (BA39), right inferior occipital gyrus (BA18), and left lingual gyrus (BA18) during imagined slipping. Significant differences in imagined slipping versus imagined walking post-training: ↑ activation of bilateral anterior cerebellum, bilateral posterior cerebellum, superior and middle temporal gyrus, right middle frontal gyrus (BA10), left SMA (BA6), left precuneus (BA31), anterior cingulate (BA25), and posterior cingulate (BA23), and left parahippocampal gyrus in imagined slipping compared to imagined walking.
Peterson et al., 2016 [77]	DTI-Connectivity	Multiple Sclerosis (n = 19)	In-place	Posterior support-surface translations (forward falls) at 4 amplitudes (3.6 cm, 6.0 cm, 8.4 cm, 12 cm)	↓ Brainstem (PPN) structural connectivity was related to antagonist (tibialis anterior) and, albeit to a lesser degree, and non-significantly (*p* = 0.07) degree, agonist (medial gastroc) onset latencies.	No correlation was observed between onset latencies and “cortical proprioceptive pathways”, which include BA3 down to the thalamus
Ragothaman et al., 2022 [74]	Whole Brain Volumes	Parkinson’s Disease (n = 96) Healthy Older Adults (n = 50)	Stepping (Push and Release Test)	Posterior (step height)	↑ Reactive step height was related to ↑ brainstem, cerebellar, and parietal brain volumes in healthy older adults.	No observed correlations in people with Parkinson’s disease
Ragothaman et al., 2022 [15]	Resting state fMRI	Parkinson’s Disease (n = 65) Healthy Older Adults (n = 42)	Stepping (Push and Release Test)	Posterior (step length)	↑ Reactive stepping was related to ↑ cerebellar-visual and ↑ cerebellar-auditory resting state connectivity	

fMRI, functional magnetic resonance imaging; SMA, supplementary motor area; BA, Brodmann area; DTI, diffusion tensor imaging; COM, center of mass; FA, fractional anisotropy; GM, gray matter; L, left; R, right; PPN, pedunculopontine nucleus; MD, mean diffusivity; MA, mode of anisotropy; COP, center of pressure; EEG, electroencephalography; FC, functional connectivity; cm, centimeters; cm/s, centimeters per second; ms, milliseconds; PET, positron emission tomography;↑, increased;↓, decreased.

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
