# Peer review of "The Neural Contributions to Reactive Balance Control: A Scoping Review of EEG, fNIRS, MRI, and PET Studies"

_brainsci, 2025, doi:10.3390/brainsci15121330_

Round 1
Reviewer 1 Report
Comments and Suggestions for Authors
This scoping review examines the neuronal correlates of reactive balance control across different populations. The article is well-articulated and makes significant contributions to literature. I would like to offer several observations concerning its overall scope.
- This review largely replicates Purohit 2022, but incorporates additional studies and broadens its focus to include neuroimaging methods such as MR and PET. The introduction provides a strong justification for this new review. However, it would be helpful to discuss what unique contributions and findings this review offers beyond previous similar reviews.
- The primary aim of this review is to better understand the neural mechanisms behind falls and improvements in reactive balance. The study populations were quite diverse, with most research conducted on healthy young adults. The conclusion should address what these results imply for healthy young adults, not just for aging or neurological populations.
- In the materials and methods section, inclusion and exclusion criteria for articles were not mentioned.
- The methodologies in the MR studies varied significantly; some were cross-sectional, while others were interventional. Since inclusion criteria were not specified, it's unclear whether interventional studies should have been included. Additionally, the interventions themselves are not clearly described in Table 3. If EEG and fNIRS studies were exclusively observational, then MR and PET studies should also perhaps be limited to observational types.
Author Response
Reviewer #1 Comments:
This scoping review examines the neuronal correlates of reactive balance control across different populations. The article is well-articulated and makes significant contributions to literature. I would like to offer several observations concerning its overall scope.
- This review largely replicates Purohit 2022, but incorporates additional studies and broadens its focus to include neuroimaging methods such as MR and PET. The introduction provides a strong justification for this new review. However, it would be helpful to discuss what unique contributions and findings this review offers beyond previous similar reviews.
Response: We have added several sentences to the first paragraph of the discussion to address how the findings of this review expand upon and offer unique contributions from the Purohit and Bhatt review.
- The primary aim of this review is to better understand the neural mechanisms behind falls and improvements in reactive balance. The study populations were quite diverse, with most research conducted on healthy young adults. The conclusion should address what these results imply for healthy young adults, not just for aging or neurological populations.
Response: The conclusion has been redrafted and includes several sentences that address this comment specifically.
- In the materials and methods section, inclusion and exclusion criteria for articles were not mentioned.
Response: We have now clarified the inclusion and exclusion criteria. See narrative added to the end of the 3rd paragraph of the methods section.
- The methodologies in the MR studies varied significantly; some were cross-sectional, while others were interventional. Since inclusion criteria were not specified, it's unclear whether interventional studies should have been included. Additionally, the interventions themselves are not clearly described in Table 3. If EEG and fNIRS studies were exclusively observational, then MR and PET studies should also perhaps be limited to observational types.
Response: All study types were included; however, data from interventional studies only included baseline data. Due to this, we have removed one study (Monaghan et. al., 2024), which did not include baseline imaging data. This is clarified in the inclusion criteria (3rd paragraph of the methods section).
Reviewer 2 Report
Comments and Suggestions for Authors
In this scoping review article, “The neural contributions to reactive balance control: a scoping review,” authored by Monaghan et al. and submitted to Brain Sciences, the authors sought to identify the neural pathways involved in reactive balance control. Although it is a very interesting and informative review that could prove a useful addition to scientific knowledge for both the scientific community and general readers, some portions should be revised before publication.
- The abstract can be improved through a complete revision.
- It is suggested that the title of this review be revised to make it clearer and more understandable.
- It is suggested to address all such issues that are frequent in the whole text such as “Increased activation of the prefrontal cortex (PFC) was consistently observed with perturbations [14, 67, 68, 70-73], particularly noted during the first several perturbations and diminishing over time, with the exception of orbitofrontal activation which remained constant [67]”.
- Under the heading 3.2.2. Influence of aging and neurologic injury or disease for fNIRS analyses, authors say, “one study reported and then in one study”. It is suggested to revise the whole draft.
- More figures can be added that may interest readers.
- In the tables, reference studies can be arranged in chronological order.
- The table font style is different from the other text.
- Table 1 can be improved by merging the Main findings into one column instead of two.
- The title of all should be simplified; there is no need for adding a long text such as “Studies included in this review that utilized electroencephalography (EEG)”.
- Heading 1.1. EEG correlates of reactive balance it can be started in another way.
- Heading 3.1.2. Methodologic approaches using EEG, should be rephrased.
- It is noted that the same pattern for naming headings has not been followed, which gives a negative impression of the quality of the review. It is suggested to take notice of this issue.
Author Response
Reviewer #2 comments:
In this scoping review article, “The neural contributions to reactive balance control: a scoping review,” authored by Monaghan et al. and submitted to Brain Sciences, the authors sought to identify the neural pathways involved in reactive balance control. Although it is a very interesting and informative review that could prove a useful addition to scientific knowledge for both the scientific community and general readers, some portions should be revised before publication
- The abstract can be improved through a complete revision.
Response: We have redrafted the abstract.
- It is suggested that the title of this review be revised to make it clearer and more understandable.
Response: We have revised the title to “The neural contributions to reactive balance control: a scoping review of EEG, fNIRS, MRI, and PET studies”.
- It is suggested to address all such issues that are frequent in the whole text such as “Increased activation of the prefrontal cortex (PFC) was consistently observed with perturbations [14, 67, 68, 70-73], particularly noted during the first several perturbations and diminishing over time, with the exception of orbitofrontal activation which remained constant [67]”.
Response: This sentence has been restructured and rephrased as follows: “Across studies, perturbations consistently elicited increased activation in prefrontal regions [14, 67, 68, 70–73]. Several studies reported that this activation was strong-est during the initial perturbations and diminished with repeated exposures. An exception was the orbitofrontal cortex, where activation remained stable over time [67].” Additionally, the manuscript has been reviewed to identify and rephrase similar sentences with similar structures.
- Under the heading 3.2.2. Influence of aging and neurologic injury or disease for fNIRS analyses, authors say, “one study reported and then in one study”. It is suggested to revise the whole draft.
Response: We have revised this section of the manuscript to reduce repetitive narrative structure and improve the readability of this section.
- More figures can be added that may interest readers.
Response: While we appreciate this suggestion, we feel that figure 2 adequately represents the findings of this review and feel that additional figures would prove redundant. Should this reviewer wish for a specific figure we would be willing to consider a specific request.
- In the tables, reference studies can be arranged in chronological order.
Response: While we appreciate this suggestion, we have opted to organize the studies in alphabetical order by first author. Using this approach can keep studies from the same authors next to each other which is particularly helpful to readers when the studies share very similar structures.
- The table font style is different from the other text.
Response: This has been modified to match the narrative text.
- Table 1 can be improved by merging the Main findings into one column instead of two.
Response: These columns have been merged.
- The title of all should be simplified; there is no need for adding a long text such as “Studies included in this review that utilized electroencephalography (EEG)”.
Response: The table headings have been simplified.
- Heading 1.1. EEG correlates of reactive balance it can be started in another way.
Response: This has been rephrased.
- Heading 3.1.2. Methodologic approaches using EEG, should be rephrased.
Response: This has been rephrased.
- It is noted that the same pattern for naming headings has not been followed, which gives a negative impression of the quality of the review. It is suggested to take notice of this issue.
Response: We have reviewed and modified the headings to ensure naming consistency throughout the review. Of note, there are a few minor deviations which are due to differences necessitated by imaging modality and relevance for appropriate organization of reporting.
Reviewer 3 Report
Comments and Suggestions for Authors
INTRODUCTION
This is enlightening, starting with falls, concepts, reactive equilibrium, and ending with the systematic review that guided the study.
METHODOLOGY
Everything is correct; the appendix shows the search.
RESULTS
Tables need improvement for easier visualization; the font needs to be changed;
The subtopic structure is a bit complex; it wasn't explained in the methods; a paragraph is needed in the methodology explaining how the results will be presented.
The objectives don't address this topic structure (by type of investigation: EEG, fNIRS...), this needs to be included in the objectives.
I reiterate, the tables are very difficult to visualize. This needs improvement.
DISCUSSION
Figure 2 is enlightening, congratulations.
The discussion is organized according to the objectives and guiding questions.
I suggest reviewing the abbreviations throughout the text; I suggest writing the study's limitations as section 4.7.
I suggest developing the conclusions further; the conclusion does not address all the results.
Author Response
Reviewer #3 comments:
INTRODUCTION:
This is enlightening, starting with falls, concepts, reactive equilibrium, and ending with the systematic review that guided the study.
METHODOLOGY:
Everything is correct; the appendix shows the search.
RESULTS:
- Tables need improvement for easier visualization;
Response: We agree. We have modified the tables to try to improve their readability. As an alternative, we have provided the tables in wide format which maybe for easily visualized and reviewed by readers.
- the font needs to be changed;
Response: This has been fixed.
- The subtopic structure is a bit complex; it wasn't explained in the methods; a paragraph is needed in the methodology explaining how the results will be presented.
Response: We have add several sentences to the last paragraph in the methods section to provide more explanation for the subtopic structure.
- The objectives don't address this topic structure (by type of investigation: EEG, fNIRS...), this needs to be included in the objectives.
Response: A sentence has been added to the last paragraph to the introduction to provide clarity to this point.
- I reiterate, the tables are very difficult to visualize. This needs improvement.
Response: See response #1.
DISCUSSION:
- Figure 2 is enlightening, congratulations
Response: Thank you!
- The discussion is organized according to the objectives and guiding questions.
I suggest reviewing the abbreviations throughout the text;
Response: All abbreviations have been defined throughout the text and in table legends if they are used in tables. We have also reviewed the manuscript for abbreviations that are used infrequently and removed these abbreviations in order to reduce the use of abbreviations.
- I suggest writing the study's limitations as section 4.7.
Response: The studies limitations are present in section 4.6.
- I suggest developing the conclusions further; the conclusion does not address all the results.
Response: The conclusion has been redrafted to more comprehensively address the breadth of the results.